# Inhibiting WNT Ligand Production for Improved Immune Recognition in the Ovarian Tumor Microenvironment

**DOI:** 10.3390/cancers12030766

**Published:** 2020-03-24

**Authors:** Whitney N. Goldsberry, Selene Meza-Perez, Angelina I. Londoño, Ashwini A. Katre, Bryan T. Mott, Brandon M. Roane, Nidhi Goel, Jaclyn A. Wall, Sara J. Cooper, Lyse A. Norian, Troy D. Randall, Michael J. Birrer, Rebecca C. Arend

**Affiliations:** 1Department of Obstetrics and Gynecology, University of Alabama at Birmingham, Birmingham, AL 35294, USA; alondono@uabmc.edu (A.I.L.); akatre@uabmc.edu (A.A.K.); bmott12@uab.edu (B.T.M.); broane@uabmc.edu (B.M.R.); nzg0014@uab.edu (N.G.); jarquiette@uabmc.edu (J.A.W.); mbirrer@uabmc.edu (M.J.B.); rarend@uabmc.edu (R.C.A.); 2Division of Immunology & Rheumatology, University of Alabama at Birmingham, Birmingham, AL 35294, USA; selenemezaperez@uabmc.edu (S.M.-P.); troyrandall@uabmc.edu (T.D.R.); 3HudsonAlpha Institute for Biotechnology, Huntsville, AL 35806, USA; sjcooper@hudsonalpha.org; 4Department of Nutrition Sciences, University of Alabama at Birmingham, Birmingham, AL 35294, USA; lnorian@uab.edu; 5O’Neal Comprehensive Cancer Center, University of Alabama at Birmingham, Birmingham, AL 35294, USA

**Keywords:** Wnt, Wnt inhibition, ovarian cancer, tumor microenvironment, ID8 cells, ID8p53^−/−^ cells, murine ovarian cancer, CGX1321, PORCN inhibitor

## Abstract

In ovarian cancer, upregulation of the Wnt/β–catenin pathway leads to chemoresistance and correlates with T cell exclusion from the tumor microenvironment (TME). Our objectives were to validate these findings in an independent cohort of ovarian cancer subjects and determine whether inhibiting the Wnt pathway in a syngeneic ovarian cancer murine model could create a more T-cell-inflamed TME, which would lead to decreased tumor growth and improved survival. We preformed RNA sequencing in a cohort of human high grade serous ovarian carcinoma subjects. We used CGX1321, an inhibitor to the porcupine (PORCN) enzyme that is necessary for secretion of WNT ligand, in mice with established ID8 tumors, a murine ovarian cancer cell line. In order to investigate the effect of decreased Wnt/β–catenin pathway activity in the dendritic cells (DCs), we injected ID8 cells in mice that lacked β–catenin specifically in DCs. Furthermore, to understand how much the effects of blocking the Wnt/β–catenin pathway are dependent on CD8^+^ T cells, we injected ID8 cells into mice with CD8^+^ T cell depletion. We confirmed a negative correlation between Wnt activity and T cell signature in our cohort. Decreasing WNT ligand production resulted in increases in T cell, macrophage and dendritic cell functions, decreased tumor burden and improved survival. Reduced tumor growth was found in mice that lacked β–catenin specifically in DCs. When CD8^+^ T cells were depleted, CGX1321 treatment did not have the same magnitude of effect on tumor growth. Our investigation confirmed an increase in Wnt activity correlated with a decreased T-cell-inflamed environment; a relationship that was further supported in our pre-clinical model that suggests inhibiting the Wnt/β–catenin pathway was associated with decreased tumor growth and improved survival via a partial dependence on CD8^+^ T cells.

## 1. Introduction

Ovarian cancer persists as the deadliest gynecologic malignancy in the United States. In 2019, there were expected to be 22,530 new cases, with 13,980 predicted deaths from the disease [1]. Research to date indicates that immunotherapies in ovarian cancer have been disappointing because immunotherapies, such as check-point inhibitors, require a “hot,” or T-cell-inflamed tumor microenvironment (TME), and most ovarian cancers have a “cold,” or non-T-cell-inflamed TME [2,3]. In order to improve the poor clinical response rates to immunotherapies, there is an urgent need to identify strategies that cause the TME to become “hotter,” by increasing T cell infiltration or decreasing T cell exclusion [4,5].

The immune-cancer landscape is complex. Numerous factors work to either promote or limit protective anti-tumor immunity. Immune-mediated control of tumor growth is dependent on not only the immune cells’ ability to infiltrate the tumor as well as the ability to remain active in this hostile environment [6]. When tumor-infiltrating lymphocytes (TILs) remain present and activated this creates a hot TME, which has been correlated with improved prognosis and with increased sensitivity to chemotherapeutic treatment in high grade serous ovarian carcinoma (HGSC) [7,8]. An immunologically cold TME may lack effector TILs and/or contain increased frequencies of CD4^+^ T regulatory cells (Tregs). Tregs act to inhibit the function of cytotoxic T cells (CTL), and thus are tumor-promoting and lead to decreased chemotherapy sensitivity [9]. High levels of CD4^+^ Tregs may lead to the exclusion of CD8^+^ T cells, and thus the CD8^+^/Treg ratio has been shown to be predictive of overall survival in ovarian cancer [10]. While CD4^+^ and CD8^+^ T cells are important factors in the TME, there are many additional immune cells, such as macrophages and dendritic cells (DCs) that are critical in determining the ability of the immune system to control tumor progression.

Previous analysis of HGSC tumors by The Cancer Genome Atlas (TCGA) consortium demonstrated a strong negative correlation between expression signatures associated with Wnt activity and T cell infiltration. Thus, ovarian cancer tumors with evidence of robust Wnt signaling had an immunologically cold TME [3]. This finding is consistent with prior studies supporting the Wnt/β-catenin pathway as a poor prognostic marker in advanced ovarian cancer patients [11]. Immunohistochemistry analysis confirmed that accumulation of β-catenin, a key Wnt signaling mediator, in epithelial ovarian cancer samples was associated with a decrease in progression-free survival (PFS) and resistance to platinum-based chemotherapy [11]. This relationship is consistent with well-established data that have shown that aberrations in the components of the Wnt/β-catenin pathway, for example, alterations in the β-catenin degradation complex, are associated with multiple cancers. The β-catenin degradation complex consists of Axin, adenomatous polyposis coli (APC), casein kinase 1α (CK1α) and glycogen synthase kinase 3β (GSK3β). Genetic mutations in the components of this complex, such as *APC* and *AXIN2*, are well known to promote colorectal carcinomas [12,13,14,15]. *AXIN1* mutations were found in hepatocellular carcinomas [16]. Deletions of *GSK3B* have been associated with progression of acute myeloid leukemia and stabilization of β-catenin in melanoma [17,18]. Although many of these links have been observed in ovarian cancer as well (reviewed in [19]), the mechanism by which Wnt signaling in the TME influences TILs in ovarian carcinoma is largely unknown.

Additional studies have identified the Wnt/β-catenin pathway as a regulator of the TME [20,21]. β-catenin stabilization has been associated with increased Treg survival [22]. Induction of the Wnt/β-catenin pathway was found to arrest CD8^+^ T cell differentiation into effector cells [23]. Alterations in the signaling cascade allowing high levels of WNT ligand to enter the TME, such as an upregulation of the transmembrane receptor frizzled (Fzd) in DC precursors, was correlated with an increased production of IL-10 and IL-12, leading to irregularities in CTL priming by DCs [24,25,26,27,28]. Furthermore, deletions of β-catenin in DCs were found to reduce Treg responses and delay tumor growth [24]. Additionally, DC-specific deletions of the Fzd co-receptor low-density lipoprotein receptor-related protein-5/6 (LRP5/6) in a murine tumor model showed delayed tumor growth with enhanced effector T cell differentiation and decreased Treg differentiation, which was then mimicked pharmacologically via use of the PORCN inhibitor IWP-L6 [29]. These findings indicate that therapeutic intervention to reduce Wnt/β-catenin signaling may alter the TME in ovarian cancer, allowing an increase in TILs, resulting in an inflamed state, which has been shown to correlate with improved outcomes.

A variety of Wnt inhibitors are currently being explored in various clinical trials. CGX1321 is an orally bioavailable small molecule that inhibits Wnt signaling by blocking Wnt ligand secretion, specifically by inhibiting PORCN in the endoplasmic reticulum [30]. Wnt ligands are modified by PORCN thru an attachment of palmitoleic acid prior to extracellular exportation [30]. Once the lipid attachment is completed, Wnt protein is then transported to the plasma membrane for secretion (Figure 1) [31]. CGX1321 is currently in two phase 1 trials for advanced gastrointestinal tumors and solid tumors, with or without pembrolizumab (NCT035007998, NCT02675946). Based on the data described and our previous work, we suggest that CGX1321 may also be of benefit in ovarian cancer patients.

Our goal was to utilize inhibition of the Wnt/β-catenin pathway in order to convert the tumor milieu from an immunologically suppressed state to an inflamed state, which could ultimately improve the potential for immunotherapeutic intervention in HGSC. We first sought to validate previously reported negative correlations between Wnt signaling and a hot TME immunological status in our own cohort of ovarian cancer patients. Next, we asked if targeting the Wnt pathway via use of a PORCN inhibitor would reverse the phenomena of T cell exclusion and promote a hot TME in a clinically relevant pre-clinical murine model of ovarian cancer. For these studies we used the ID8 parental cell line [32,33]. Interestingly, in this cell line, unlike human ovarian cancer, there are no deleterious mutations found in the *Trp53* gene, which is known to be commonly mutated in 97% of human HGSC [32,34]. Therefore, in an attempt to more closely mimic human ovarian HGSC, we also performed studies using a p53-deficient ID8 cell line (ID8p53^−/−^).

## 2. Results

### 2.1. Wnt and Immune Gene Signatures in Ovarian Cancer Patient Samples

RNA sequencing (RNA-seq) on 57 untreated human high grade serous ovarian cancer (HGSC) tissue samples was used to calculate Wnt activity and T cell infiltration using previously published signatures (Appendix A) [3,35]. Higher Wnt gene expression correlated with lower T cell expression, referred to as a “cold” T cell signature. Treatment naïve tumor samples with increased expression of the T cell signature genes, referred to as “hot” tumors, were found to have significantly lower relative Wnt activity, *p* value = 0.002 (Figure 2a). A subset (*n* = 17) of the 57 treatment naïve samples had matched samples after receiving neo-adjuvant chemotherapy (NACT). Changes in Wnt activity were calculated for these samples (Appendix A). Cold tumors and tumors with activating Wnt mutations and/or higher Wnt/β-catenin pathway activity have been shown to have a poorer prognosis, thus we sought to investigate whether the influence of chemotherapy changed either of these and if so, how this correlated with patient outcomes (Appendix A) [4,7,11]. Kaplan–Meier survival curves, including PFS and overall survival (OS), were calculated in relation to decrease in Wnt activity (Figure 2a,b). While results were not statistically significant for OS, both curves were suggestive of improved patient outcome with decreased Wnt activity (PFS *p* value = 0.039, OS *p* value = 0.220). Four of the 17 patients had a decrease in Wnt activity >10%, 3 of which had a hot signature originally. T cell signature or the level of Wnt pathway gene expression did not change in the majority of patients after NACT (Figure 2c). In tumors where the fold change of Wnt activity was the greatest at the time of IDS, there was a trend towards increased PFS (Figure 2c).

### 2.2. In Vivo Inhibition of Wnt Signaling Using CGX1321

Prior studies indicate that the ID8p53^−/−^ line has more aggressive tumor outgrowth, ascites formation, higher percentages of myeloid cells in the TME and decreased animal survival than the parental ID8 line [34]. Thus, we performed pre-clinical murine experiments using both tumor lines. Fully immune competent C57Bl/6 mice were injected intraperitoneally with ID8 or ID8p53^−/−^ cells, on day 0 (*n* = 7). Starting on day 28 post-tumor challenge, mice were treated orally with the PORCN inhibitor CGX1321 daily for 14 days (Figure 3a). Survival was monitored until animals were moribund. No significant difference in survival was observed in mice treated with CGX1321 in the ID8p53^−/−^ mouse model (Figure 3b). However, we found significantly increased survival in mice treated with CGX1321 in the parental ID8 mouse model, *p* value = < 0.001 (Figure 3b). This experiment was repeated (*n* = 5), with consistent results. In a separate animal study, (*n* = 10) mice were sacrificed after completion of CGX1321 treatment on day 42 of tumor challenge. Tumor burden was analyzed by omentum weight. Omentum tumors in the ID8 model were statistically smaller compared to vehicle control, *p* value = 0.006. In contrast, we found only a trending decrease in omentum weight in the ID8p53^−/−^ mouse experimental model, *p* value = 0.070 (Figure 3c).

### 2.3. Characterization of the Wnt Profile in ID8 and ID8p53^−/−^ Cell Lines

To better understand the baseline tumor proliferation differences between ID8 and ID8p53^−/−^ cell lines, we intraperitoneally injected C57Bl/6 mice, without treatment, for either survival analysis or sacrifice on tumor challenge day 42 for tumor analysis. We observed a statistically significant increase in survival with ID8, compared to ID8p53^−/−^, *p* value = < 0.001 (Appendix A). On tumor challenge day 42, omenta weights were compared between the two cell lines. ID8p53^−/−^ omentum tumors were significantly larger than ID8 omentum tumors (Appendix A, *p* value = 0.001). Histologic comparison of these omenta also revealed a near complete replacement of omentum fat by tumor in ID8p53^−/−^ injected mice (Appendix A). Western blot analysis on ID8 and ID8p53^−/−^ cell line lysates (Figure 4a, Appendix A) revealed a decreased baseline level of β–catenin in ID8p53^−/−^, verified via densitometry analysis (Figure 4a). RNA-seq of omentum tumor from ID8 or ID8p53^−/−^ in C57Bl/6 mice showed reduced transcript levels for several Wnt pathway genes in the p53 knockout line: *WNT2B*, *WNT5A*, *FZD4* and *AXIN2* (Figure 4b). These findings imply an increased aggressive nature of ID8p53^−/−^ tumors, albeit a decreased dependence on the Wnt/β-catenin signaling pathway, correlating to the lack of robust response with CGX1321 treatment in ID8p53^−/−^ tumors.

### 2.4. In Vivo Effect of Wnt Signaling Inhibition in ID8 Omentum Tumor and Microenvironment

The NanoString Pan Immune profile was used to analyze RNA expression of omentum tumor in the ID8 mouse model following CGX1321 treatment. Gene signatures of CGX1321 treated mice compared to the vehicle control illustrated an increase in gene expression related to T cell functions, macrophage functions, DC functions and antigen processing (Figure 5a, Appendix A). Analysis of leukocyte populations from the tumor bearing-omentum by flow cytometry at day 42 post-ID8 tumor challenge showed an increase in macrophages and DCs with CGX1321 treatment, *p* value = 0.016 and 0.008. CD4^+^ Treg and CD8^+^ T cell frequencies were unchanged by CGX1321 treatment at this timepoint (day 42), *p* value = 0.534 and *p* value = 0.051 (Appendix A). Flow cytometry gating for DCs and macrophages in omentum tumor are displayed in Appendix A, followed with Treg and CD8^+^ T cell gating strategies for omentum tumor in Appendix A.

### 2.5. Inhibition of β–Catenin Signaling in Dendritic Cells Decreases Tumor Burden

Increased Wnt signaling is thought to increase the regulatory state of DCs, with increased IL-10 and IL-12 production of DCs (Figure 6a) [19]. To test the importance of Wnt signaling in DCs, we generated a mouse model in which β–catenin was knocked out specifically in DCs, by crossing CD11c-cre mice with β–catenin^−/−fl/fl^ mice (CD11c-cre × β–catenin^−/−fl/fl^) (Figure 6a). At day 42, post-ID8 tumor challenge in CD11c-cre × β–catenin^−/−fl/fl^ mice, omentum tumor weights were decreased compared to control β–catenin^−/−fl/fl^ mice, *p* value = 0.101, without treatment (Figure 6b). Treatment with CGX1321 resulted in a significant decrease in omentum weight, *p* value = 0.045 (Figure 6b). The flow cytometric analysis of leukocytes within omentum tumors revealed an increase in CD8^+^ T cells in CD11c-cre × β–catenin^−/−fl/fl^ mice, *p* value = 0.001 (Figure 6c). In concordance with decreased tumor burden with CGX1321 treatment, there was also a significant increase in CD8^+^ T cellular percentage seen in tumors, *p* value = <0.001 (Figure 6c). The loss of intrinsic DC β–catenin may promote cross-priming of CTLs, as suggested by a decreased tumor burden and an increase in CD8^+^ T cells within the TME.

### 2.6. CD8^+^ T Cells are Required for Wnt Inhibition to Significantly Inhibit Tumor Progression In Vivo

Increased Wnt activity is thought to impair priming of CD8^+^ T cells by DCs (Figure 7a). To better understand the role of CD8^+^ T cells in our syngeneic mouse model, C57Bl/6 mice were injected with ID8 tumor cells intraperitoneally. Mice were then treated, starting on tumor challenge day 28, with CGX1321 for 14 days (*n* = 10) or remained as untreated controls (*n* = 10). A subset of these mice was depleted of CD8^+^ T cells by using an anti-CD8^+^β antibody, administered on post-tumor challenge days 27, 28, 30 and 35, as shown in Figure 7b. We observed no significant decrease in omentum weight at day 42 post-ID8 tumor challenge in mice treated with CGX1321 and anti-CD8^+^β antibody, in comparison to mice who did not receive CGX1321 or antibody, *p* value = 0.318. However, mice that were not depleted of CD8^+^ T cells showed a significant decrease in omentum weight with CGX1321 treatment (Figure 7c). Depletion of CD8^+^ T cells was confirmed in the omenta of mice treated with anti-CD8^+^β antibody by flow cytometry (Appendix A). Activated CD8^+^ T cells showed a decreased tumor burden when mice were treated with CGX1321.

## 3. Discussion

Cells of the TME play critical roles in ovarian cancer patient outcomes. It is established that an inflamed tumor milieu, rich in TILs, represents increased recognition of the tumor by the immune system, which improves overall survival and response to both chemotherapeutic agents and immunotherapies [7]. Therefore, exploration of the TME in pre-clinical models is vital to understand the influences of the immune system, both locally and systemically, as we investigate future treatments.

Previous TCGA analysis of HGSC revealed a strong inverse correlation between Wnt/β-catenin signaling pathway gene signatures and infiltrating T cell gene signatures [3]. Here, we validate this correlation using RNA-seq from an independent patient cohort of 57 pre-treatment HGSC tumor tissues (Figure 2a). From a subset of these patients, we correlated PFS and OS with Wnt activity decreases after NACT, finding a trend toward improved patient outcomes with Wnt decrease (Figure 2b,c). Additionally, PFS was inversely correlated with fold-change in Wnt signaling post-NACT, supporting the idea that increased Wnt activity leads to poor treatment sensitivity and thus worse survival (Figure 2d). Our findings further support the correlation between increased TILs with decreased Wnt activity in ovarian cancer; as such, we postulate that therapeutically inhibiting Wnt signaling could increase TILs and improve treatment sensitivity—and potentially overall survival. This supports the idea that combining chemotherapy or immunotherapy with Wnt inhibition could improve patient outcomes in ovarian cancer.

One limitation of our study is that the ID8 murine ovarian carcinoma cell line does not mimic most HGSC in that it does not contain a *Trp53* mutation, which is found almost universally in human HGSC. Therefore, we added a *Trp53* knockout, ID8p53^−/−^ to our analysis. This model is known to produce more aggressive tumors and ascites formation, in addition to higher percentages of myeloid cells in the TME, demonstrated by flow cytometry analysis [34]. Given these differences, ID8p53^−/−^ should be viewed as a newly established murine ovarian carcinoma cell line, distinct from the ID8 cell line. We confirmed prior reports, observing decreased survival in ID8p53^−/−^ tumor challenged mice compared to those receiving an equivalent dose of parental ID8 cells (Appendix A). Interestingly, treatment with the PORCN inhibitor improved survival in mice with parental ID8 tumors but did not improve survival in ID8p53^−/−^ mice (Figure 3b). Additionally, tumor burden did not significantly decrease with CGX1321 treatment in ID8p53^−/−^ mice, but did decrease in mice receiving parental ID8 tumor challenge, suggesting that ID8p53^−/−^ tumors may not be as heavily influenced by changes in the Wnt/β-catenin signaling pathway as parental ID8 tumors (Figure 3c). This idea was supported by further in vitro investigations showing a decreased total β-catenin level in ID8p53^−/−^ tumor cells and decreased expression of Wnt/β-catenin genes on ID8p53^−/−^ omentum tumors (Figure 4a,b). Given these profound differences in the role of the Wnt/β-catenin pathway between these cell lines, all further experiments in this study were limited to the ID8 cell line. We recognize these data with ID8p53^−/−^ cells, a line thought to mimic *Trp53* mutated human HGSC, make the clinical implication of these results difficult to interpret. This mutation may influence alterations of signaling pathways in this murine ovarian cancer model that are not reflected in human HGSC. Further investigation is warranted in additional cells lines, although this was beyond the scope of our current study.

To further investigate the TME differences in ID8 omentum tumor with and without CGX1321, we used targeted gene expression measurements by NanoString. We found an increase in gene signatures related to T cell functions, macrophage functions, DC functions and antigen processing after CGX1321 treatment (Figure 5a). These findings support the relationship of decreased Wnt/β-catenin signaling associated with increased TIL function in the TME and support the idea that targeting the Wnt/β-catenin pathway may convert the tumor milieu from a “cold” TME to a “hot,” T-cell-inflamed, environment thereby promoting tumor recognition by the immune system. Further cellular analyses via flow cytometry of omentum tumor showed an increase in macrophages and DCs following CGX1321 treatment (Figure 5b). Interestingly, CD4^+^ Treg and CD8^+^ T cell frequencies at day 42 were not statistically altered in the TME after CGX1321 treatment, despite our gene expression data indicating increased T cell function (Appendix A). This discrepancy may reflect the timing of tumor harvest and sample processing. While intact cells may have escaped detection, gene signature was preserved. Future experimentation will clarify these results. Nonetheless, these findings support that decreased Wnt/β-catenin signaling may rely on both the increased presence and function of macrophages and DCs with a partial reliance on increased T cell function, converting the tumor milieu to one with improved tumor recognition.

In our mouse model with complete absence of β–catenin in DCs, there was a correlation with decreased tumor burden compared to a control model with β–catenin present in DCs. This decreased tumor burden in the β–catenin absent DC model was further enhanced with CGX1321 treatment (Figure 6b). These findings support the idea that decreased β–catenin in these antigen-presenting cells may improve tumor recognition; however, further tumor recognition occurred when the entirety of the TME had a decrease in Wnt/β–catenin signaling. These findings are consistent with prior studies where DC-specific lipoprotein receptor-related protein-5/6 (LRP5/6, a co-receptor for Wnt ligands) deletions were explored in a murine tumor model in which results showed delayed tumor growth with enhanced effector T cell differentiation and decreased Treg differentiation [29]. Interestingly, flow cytometry on these omentum tumors with DC β–catenin elimination revealed an increase in CD8^+^ T cells, a result that was further exaggerated with CGX1321 treatment (Figure 6c). This suggests tumor recognition may be additionally reliant on the improved abundance and effectiveness of CTL priming by DCs lacking β–catenin, but improved tumor recognition with CGX1321 may be partially reliant on CD8^+^ T cells.

Reliance on CD8^+^ T cells in the TME for tumor recognition was further explored by eliminating CD8^+^ T cells in tumor challenged mice with anti-CD8^+^β antibody, both with and without CGX1321 treatment. As expected, there was a decrease in tumor burden with CGX1321 treatment. However, interestingly, there was not a statically significant decrease in tumor burden when CD8^+^ T cells were depleted during CGX1321 treatment, indicating a partial reliance on CD8^+^ T cells for tumor recognition, consistent with the above genomic findings of increased T cell functions and supporting the implication of a conversion from a cold TME to one with an improved reliance on T cells for tumor recognition with a decrease in Wnt/β–catenin signaling. Future experimentation will work toward elucidating this complex role between Wnt/β–catenin signaling, antigen-presentation and CD8^+^ T cell reliance in tumor recognition further.

## 4. Materials and Methods

### 4.1. Tissue Specimen Collection and Processing

All patient studies were conducted in accordance with the ethical guidelines of the Belmont Report. IRB approval was obtained prior to human specimen collection. The ethical code is IRB-131007005. Written consent for participation in this study was obtained from all patients prior to surgery. Tissue was collected from 57 patients at the time of surgical debulking for papillary serous ovarian or primary peritoneal cancer. Tumor samples from 17 patients were collected from before and after treatment with neoadjuvant chemotherapy. All tissues were flash frozen or stored in RNA-later at the time of collection. Total RNA was extracted from tissue homogenate using the Norgen Animal Tissue RNA Purification Kit (Norgen Biotek Corporation, Thorold, ON, Canada). Tissue homogenates (FastPrep homogenizer, lysing matrix D, MP Bio, Solon, OH, USA) were treated with Proteinase K before being applied to the column, and on-column DNAse treatment was performed according to the manufacturer’s instructions. Total RNA was eluted from the columns and quantified using the Qubit RNA Assay Kit and the Qubit 2.0 fluorometer (Invitrogen, Smestad, Oslo, Norway).

### 4.2. Cell Lines and Culture

The ID8 murine epithelial ovarian cancer cell line was provided by Dr. Yancey Gillespie (University of Alabama at Birmingham, Birmingham, Alabama). ID8p53^−/−^ cell line is a derivative of ID8 with CRISPR/Cas9 gene editing for a single gene mutation (*Trp53^−/−^*). This cell line was provided by Dr. Iain McNeish (Wolfson Wohl Cancer Research Centre, Institute of Cancer Sciences, University of Glasgow, Glasgow, United Kingdom). Cells were maintained in DMEM (Corning) supplemented with 4% fetal bovine serum (Atlanta Biologicals, Flowery Branch, GA, USA), 1X insulin/selenium/transferrin (Invitrogen), penicillin/streptomycin and 2 mM L-glutamine. Cells were maintained in culture fewer than 10 passages from the parent stock. Experiments were performed at 70–80% confluency.

### 4.3. Mouse Studies

All animal studies were approved by the University of Alabama at Birmingham Institutional Animal Care and Use Committee (IACUC). The ethical code-IACUC-21227. C57Bl/6 mice were obtained from NCI Charles River (Wilmington, MA, USA). A Cre-flox C57Bl/6 mouse model, CD11c-cre × β–catenin^−/−fl/fl^ mice models, were provided by Dr. Troy Randall’s lab at University of Alabama at Birmingham. Experiments were performed in mice at 7–8 weeks of age. Intraperitoneal tumor challenge included abdominal injection with 7 × 10^6^ cells in 200 μL PBS and treatment was started 28 days after cell injection.

Mice were randomized to no treatment (Null), treatment with vehicle control, or treatment with CGX1321. Vehicle control consisted of 20% PEG400, 25% Solutol (20%) in purified water (weight/volume): 55% dextrose (5%) in water (volume:volume:volume). CGX1321 (Curegenix, Guangzhou, China) was prepared in suspension in the vehicle at 0.2 mg/mL. The appropriate amount of CGX1321 and 80% of the final required volume of the vehicle were combined and sonicated to a homogeneous suspension. The remainder 20% of final volume of vehicle was added to achieve the target concentration. Daily dosage was given at 1 mg/kg in 100 μL oral gavage. For CD8^+^ depletion studies, anti-mouse CD8^+^β antibody (Lyt 3.2) was purchased from BioXcell (BE0223). Antibody was administered via intraperitoneal injection on tumor challenge day 27, 28, 30 and 35.

### 4.4. Western Blot Analysis

Mouse ovarian cancer cell lines were seeded 1 million per well in 6 well plates. Following 24 h incubation, the cells were lysed in RIPA buffer supplemented with protease and phosphatase inhibitors. Protein concentrations were determined with the BCA Protein Assay Kit (Thermo Scientific, Vilnius, Lithuania, Pierce# 23225). Immunoblot analysis was carried out by standard techniques previously described [37]. Briefly, equal quantities of protein were subjected to SDS–PAGE under reducing conditions. Following transfer to immobilon-P membrane, successive incubations with Non-phospho (Active) β-catenin (Cell Signaling, Beverly, ME, USA, cat# 8814) at 4 °C overnight or anti-β-actin (Cell Signaling cat# 5125S) and horseradish peroxidase-conjugated secondary antibody (Cell Signaling) were carried out for 60–120 min at room temperature. The immunoreactive proteins were quantified using the ECL system (PerkinElmer, Hopkinton, MA, USA). Films showing immunoreactive bands were scanned on ChemiDoc XRS system (BIO-RAD, Hercules, CA, USA).

### 4.5. RNA Sequencing

RNA was extracted from tissue samples using the Norgen Total RNA extraction kit (cat. # 17200, Norgen, Thorold, ON, Canada). RNA Integrity Numbers (RIN) were measured using the BioAnalyzer for many, but not all samples. The range in RIN was 2.4–8.9, with a median of 7.85, with 5 samples having a value less than 7, Appendix A. We also used the percentage of reads aligning to the mitochondrial chromosome as a measure of RNA quality (PMID: 25577376). 800ng of total RNA was used as input to the NEBNext Ultra RNA Library Prep Kit for Illumina using the polyA selection method (E7530S and E7490S). Libraries were barcoded using NEBNext multiplex oligos for Illumina (E7335S). Samples were pooled and sequenced using an Illumina HiSeq and sequenced an average of 27M aligned reads per sample. Samples were processed using the published primary analysis tool, aRNApipe [38]. Count tables were analyzed using DESeq2 for differential expression analysis [38]. R version 3.6.1 was used for all statistical analysis of sequencing data.

### 4.6. Calculating Wnt Signaling and Immune Signatures

We calculated a score that represents Wnt and immune signatures from gene expression data. Genes used to determine T-cell signature and WNT activity are listed in Appendix A. First, we used the variantStabilizingTransformation function in the DESeq2 package to normalize all data. Next, we calculated a score for each immune cell type or Wnt signaling by determining the median normalized gene expression levels of all genes associated with the cell type or function. To calculate immune signatures, we took advantage of data from Danaher, et al. which identified transcripts whose expression are highly correlated with the presence of a variety of immune cells in ovarian tumors [35].

### 4.7. NanoString nCounter mRNA Analysis

Omentum tumors were harvested from mice at the time of sacrifice and stored in RNALater. RNA was isolated using Trizol Plus RNA Purification Kit (Invitrogen, cat. # 12183555) according to the manufacturer’s instructions. Aqueous phase solutions were transferred to a spin cartridge found in the PureLink RNA Mini Kit and processed similarly. Samples of high purity (A260/A280 > 2, A260/A230 > 1.4) were processed on the NanoString nCounter Flex system (NanoString Technologies, Seattle, WA, USA) using the mouse PanCan Immune pre-made panels. Briefly, 100 ng of purified RNA was hybridized for at least 16 h with the respective Reporter Code set and Capture Probes for each panel separately. The samples were purified and immobilized on the NanoString Prep Station and counted on the NanoString Digital Analyzer. Analysis was performed using nSolver 3.0 software (NanoString Technologies). Cluster 3.0 and Java TreeView-1.1.6r4 were used to create heat maps. Expression levels were normalized to housekeeping genes that were not discarded by the gNorm program in the advanced analysis module. As stated in the previous section, we utilized NanoString to calculate immune scores for specific immune cell types. Wnt signaling was not calculated for NanoString-only samples as the vast majority of the Wnt signature genes are not measured by NanoString analysis.

### 4.8. Flow Cytometry

Omentum tumors were harvested and placed in digestion buffer (DMEM 25Mm HEPES, 3.5% fatty acid free BSA plus 0.5 mg/mL collagenase + 70 ug/mL DNAse I) (all from Corning) and fragmented with scissors. Tumor fragments were placed in an orbital shaker at 250–300 rpm for 30 min at 37 °C and subsequently filtered through 70 µm nylon mesh, washed with staining buffer (1× PBS, 2% BCS and 2 mM EDTA) (all from Corning) and centrifuged at 800× g for 10 min. Erythrocyte lysis was done using homemade lysis buffer (0.15M NH_4_Cl, 10 mM KHCO_3_ and 0.1mM EDTA, pH 7.4) (all from Corning).

The following antibodies were used for staining: CD3 (100214CD3), CD11c (117318), anti-Mouse I-A/I-E (107620), NK1.1 (108716) CD45.2 (109824), (all from BioLegend, San Diego, CA, USA), CD4 (560782), CD25 (557192), CD8a (551162 A 1), CD45R (B220) (553093) (all from BD Biosciences, San Jose, CA, USA), CD366 (TIM3) (25-5870-80), CD279 (PD-1) (12-9985-83), CD103 (17-1031-82), Foxp3 (11-5773-82), CD152 (CTLA-4) (HMCD15201) (all from eBioscience/Thermofisher, Waltham, MA, USA) and live-dead discriminator 7AAD 7-Amino-AMD (129935) (Calbiochem/Millipore Sigma, Burlington, MA, USA). The following stains were used for macrophage identification: CD45.2, CD103, F4/80, CD11c, CD68, Ly6G, CD3, B220, CD19, NK1.1, CD11b, Ly6C, IA-IE. Fixation and permeabilization for intracellular markers including FoxP3 were performed using the FoxP3/Transcription Factor Staining Buffer Set (eBioscience/Thermofisher, Waltham, MA, USA). Intracellular cytokine staining was done using BD Biosciences Fixation/Permeabilization kit specifications. Samples were run using a BD FACS Canto II (BD Biosciences, San Jose, CA, USA), DIVA software 6.1.3. Data were analyzed with FlowJo version 9.9.6. DCs and macrophage cell population gating strategies were initially gated on lymphocytes by SSC-A vs FSC-A and to exclude debris. The population was then gated on singlets using FSC-A vs FSC-H, then CD45^+^ cells. This population was then gated on CD11b^+^ and Ly6C^−^. DCs were then discriminated with CD11c^+^. Macrophages were further gated by CD11b^+^. Treg and CD8^+^ T Cells were initially gated on lymphocytes by SSC-A vs FSC-A and to exclude debris. The population was then gated on singlets using FSC-A vs FSC-H, then on CD3^+^ cells. This population was then discriminated by T-cell populations, CD8^+^ vs CD4^+^. FoxP3^+^CD25^+^ Tregs were gated on CD3^+^CD4^+^ populations. Gating strategies are used for each mouse omentum tumor analyzed for these cell populations (Appendix A).

### 4.9. Statistical Analysis

For all mouse data, statistical analyses were performed using GraphPad Prism version 8.0 (San Diego, CA, USA) with the exception of NanoString data. Determination of statistical significance between group means was done using multiple unpaired independent t-tests with an alpha of 0.05. For determination of significance between more than two group means, one-way or two-way ANOVA were used correcting for multiple comparisons using Tukey’s multiple comparisons test. Survival analysis of human outcomes and mouse survival was performed using the Kaplan Meier method; curves were compared with the Mantel-Cox log-rank test.

### 4.10. Graphic Design

Graphics created with BioRender.com, including Graphical Abstract: Figure 1, Figure 3a, Figure 6a and Figure 7a,b.

## 5. Conclusions

Using the same gene signatures previously published by Luke, et al., we validated a correlation between increased WNT activity and decreased T-cell signature in our own cohort of patients with HGSC. In a subset of patients after receiving NACT, we also correlated an increase in PFS with decreased Wnt activity. We further sought to investigate if decreasing Wnt/β–catenin signaling through inhibition of PORCN would affect the T cell signature. Mice injected with ID8 cells were treated with CGX1321, a small molecule that inhibits the PORCN enzyme. Survival and tumor growth were measured with and without treatment. Changes in the TME following treatment were analyzed via flow cytometry and RNA expression. Treatment with CGX1321 decreased ID8 tumor burden and increased survival, but this was not statistically significant in the ID8p53^−/−^ cell line. After further investigation, we discovered that a fundamental difference between the parental and the ID8p53^−/−^ cell line is decreased β-catenin levels and RNA expression of Wnt-related genes in ID8p53^−/−^.

Using the NanoString Pan immune profile to analyze RNA expression, we showed that treatment with CGX1321 lead to an increase in T cell functions, macrophage functions, DC functions and antigen processing. To further characterize how these RNA expression changes effected immune cell populations, we performed flow cytometry analysis on specific immune cell populations. The number and percentage of macrophages and DCs increased with CGX1321 treatment.

In order to investigate directly knocking out all Wnt/β-catenin pathway function in DCs specifically, we compared tumor growth in mice that lacked β–catenin in DCs. In this mouse model, there was less tumor growth compared to mice with β–catenin intact in the DCs. When treating mice with no DC β–catenin activity with CGX1321, tumor growth was even further decreased. These results suggest that directly blocking β–catenin activity in the DCs alone decreased tumor growth, but additional PORCN inhibition, leading to decreased levels of WNT ligand overall, affects tumor growth and other cells in the TME. This was further supported in tumor analysis revealing an increase in CD8^+^ T cells in the TME in both the untreated and CGX1321 treated tumors in the DC β–catenin absent model.

While DCs are important in immune recognition and destruction of tumor cells, they do not directly kill tumor cells. They act as antigen presenting cells (APCs), presenting tumor associated antigens to cytotoxic T cells, which are the most important cells in immune mediated tumor cell death. Thus, we further investigated if CGX1321 would have the same effect on tumor growth in the absence of CD8^+^ T cells. We found that CGX1321 did not have the same degree of effect on decreased tumor burden when given with anti-CD8^+^β antibody. This supports the idea that PORCN inhibition is at least partially dependent on CD8^+^ T cells.

Given our findings, we believe that the Wnt/β–catenin signaling pathway plays an important role in sensitizing tumors to immune-mediated cell death. This could be intrinsic cell death, or cell kill, that is further augmented by immunotherapy. With this, altering other influences within the TME on TILs in order to convert the tumor milieu to a “hot” tumor for improved tumor recognition by inhibiting the Wnt/β–catenin signaling pathway may be a promising direction in drug development for ovarian cancer.

## Figures and Tables

**Figure 1 cancers-12-00766-f001:**
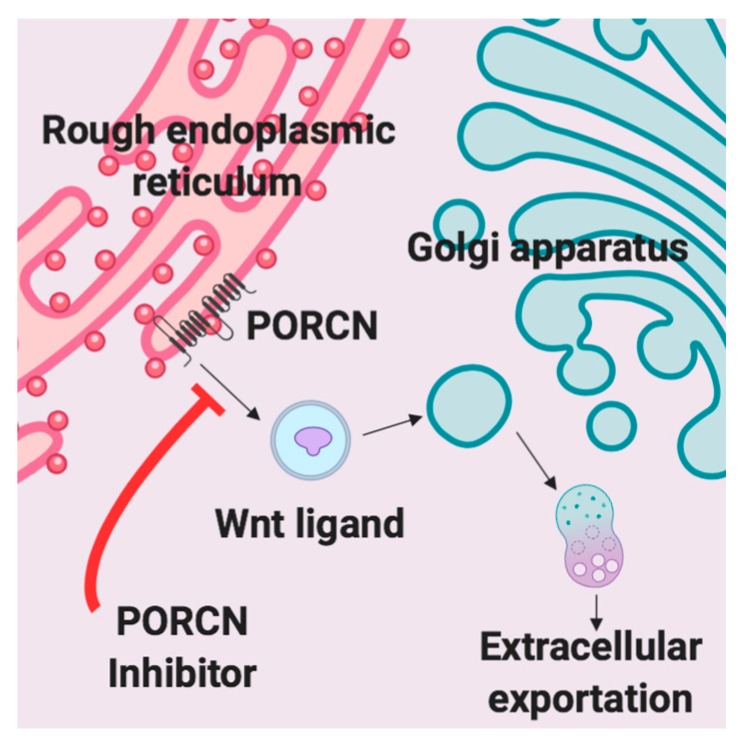
PORCN inhibition. Extracellular exportation of WNT ligand is prohibited due to a lack of a necessary lipid attachment to WNT by PORCN.

**Figure 2 cancers-12-00766-f002:**
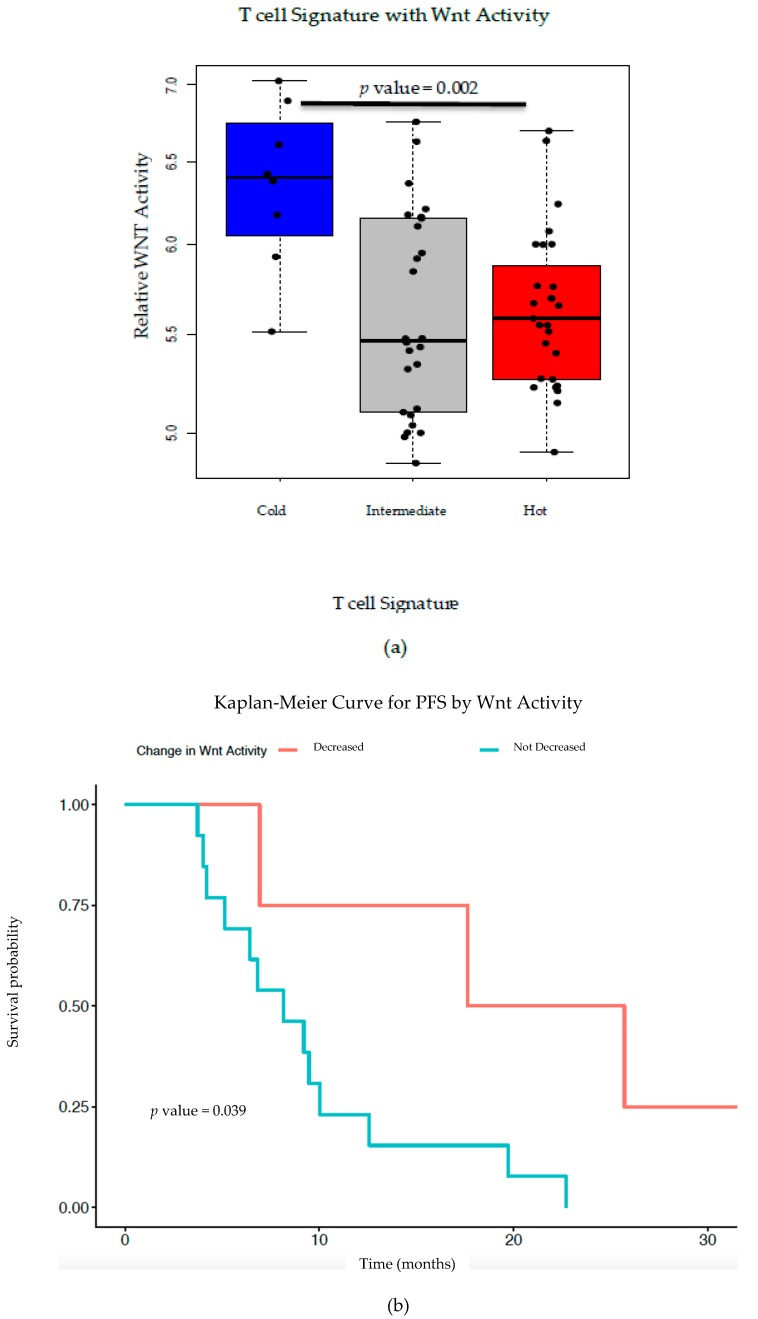
Wnt and immune gene signature in ovarian cancer patient samples. (**a**) T cell signatures and Relative Wnt Activity scores calculated from RNA-seq data for 57 treatment naïve human HGSC tissues show a negative correlation. (**b**) Kaplan–Meier curve for PFS (*n* = 17) based on decreases in Wnt activity after NACT. (**c**) Kaplan–Meier curve for OS (*n* = 17) in correlation with decreased Wnt activity after NACT. (**d**) PFS shows a negative association with fold change in Wnt activity measured by signature genes in matched post- versus pre-NACT in 17 HGSC patients. Samples are labeled based on T cell signature changes, with a solid color indicating no change, a blue square with red lines indicating cold-to-hot signature change and a red square with blue lines indicating a hot-to-cold signature change.

**Figure 3 cancers-12-00766-f003:**
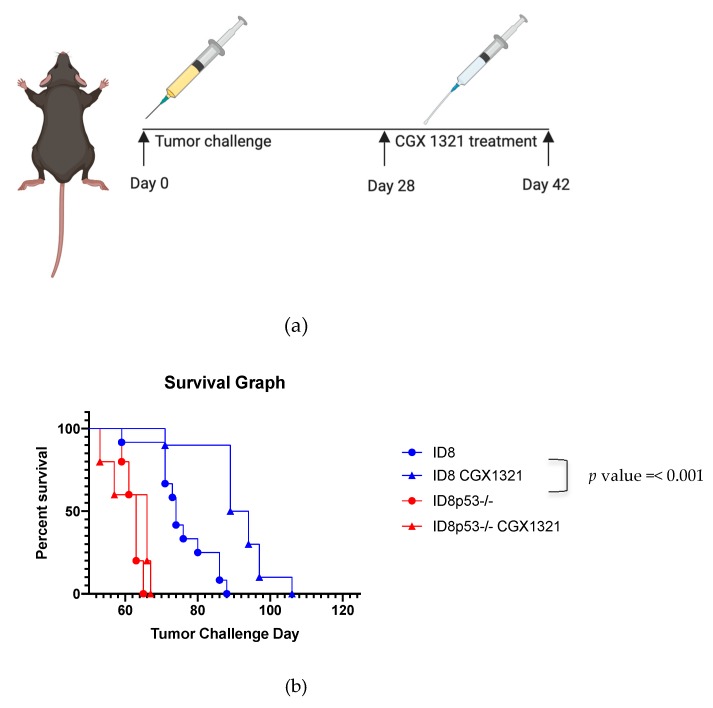
In vivo inhibition of Wnt signaling using CGX1321. (**a**) Scheme of the timeline for tumor challenge and treatment in C57Bl/6 mice for both cell lines. Mice were either then monitored for survival or sacrificed on Day 42. (**b**) Survival was increased with CGX1321 treatment in ID8 tumor challenge, but not tumor challenge with ID8p53^−/−^. (**c**) Omentum weight had a statistically significant decrease with CGX1321 treatment after 42 days of tumor challenge with ID8, but not with ID8p53^−/−^ tumor challenge.

**Figure 4 cancers-12-00766-f004:**
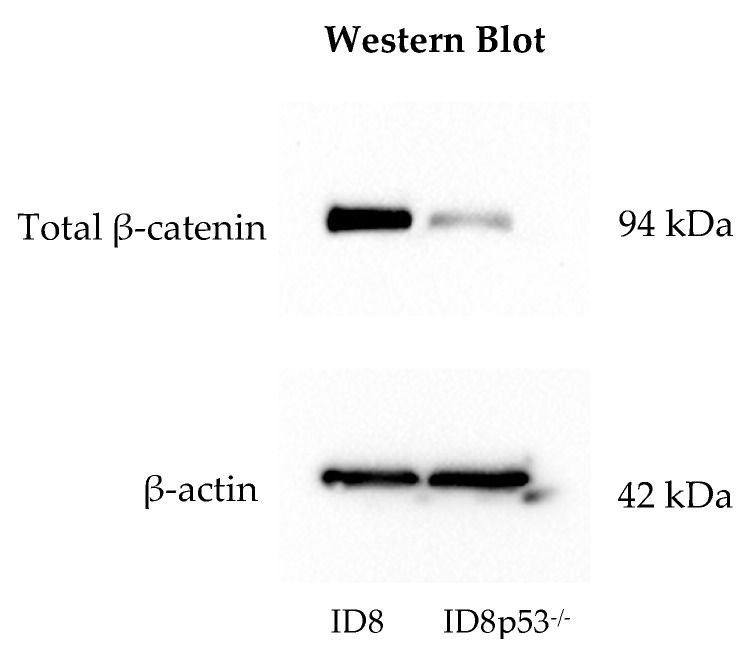
Characterization of the Wnt profile in ID8 and ID8p53^−/−^ cell lines. (**a**) Western blot with reduced baseline β–catenin levels in the ID8p53^−/−^ cell line, with densitometry analysis. (**b**) Normalized Wnt related gene expression data for *WNT2B*, *WNT5A*, *FZD4* and *AXIN2* in ID8 (*n* = 6) and ID8p53^−/−^ (*n* = 6) was decreased in the ID8p53^−/−^ cell line compared to the parental ID8 cell line.

**Figure 5 cancers-12-00766-f005:**
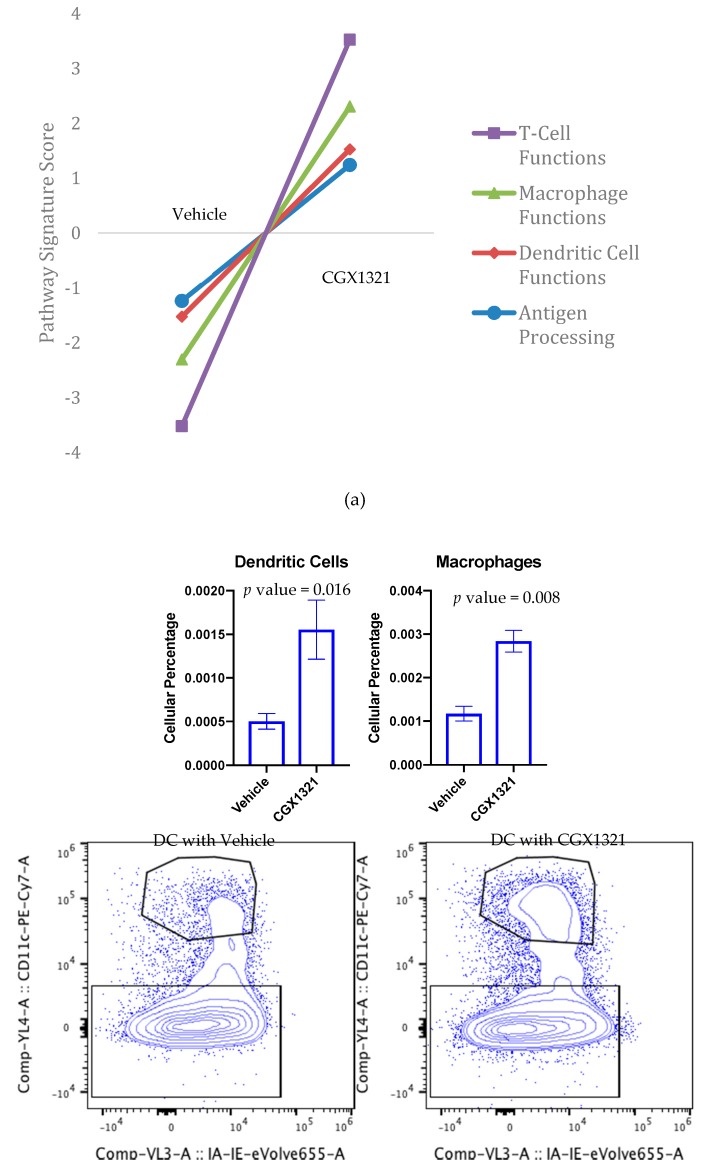
In vivo effect of Wnt signaling inhibition in omentum tumor and the microenvironment. (**a**) NanoString analysis of omentum at 42 days tumor challenge with ID8 cells showed an increase in gene signatures for T cell functions, macrophage functions, dendritic cell functions and antigen processing with CGX1321 treatment. (**b**) After 42 days of ID8 tumor challenge, flow cytometry of omentum tumor showed an increase in macrophages and DCs with CGX1321 treatment. Flow gating strategy is shown first for omentum tumor DCs with vehicle or CGX1321 treatment, and similarly for macrophages.

**Figure 6 cancers-12-00766-f006:**
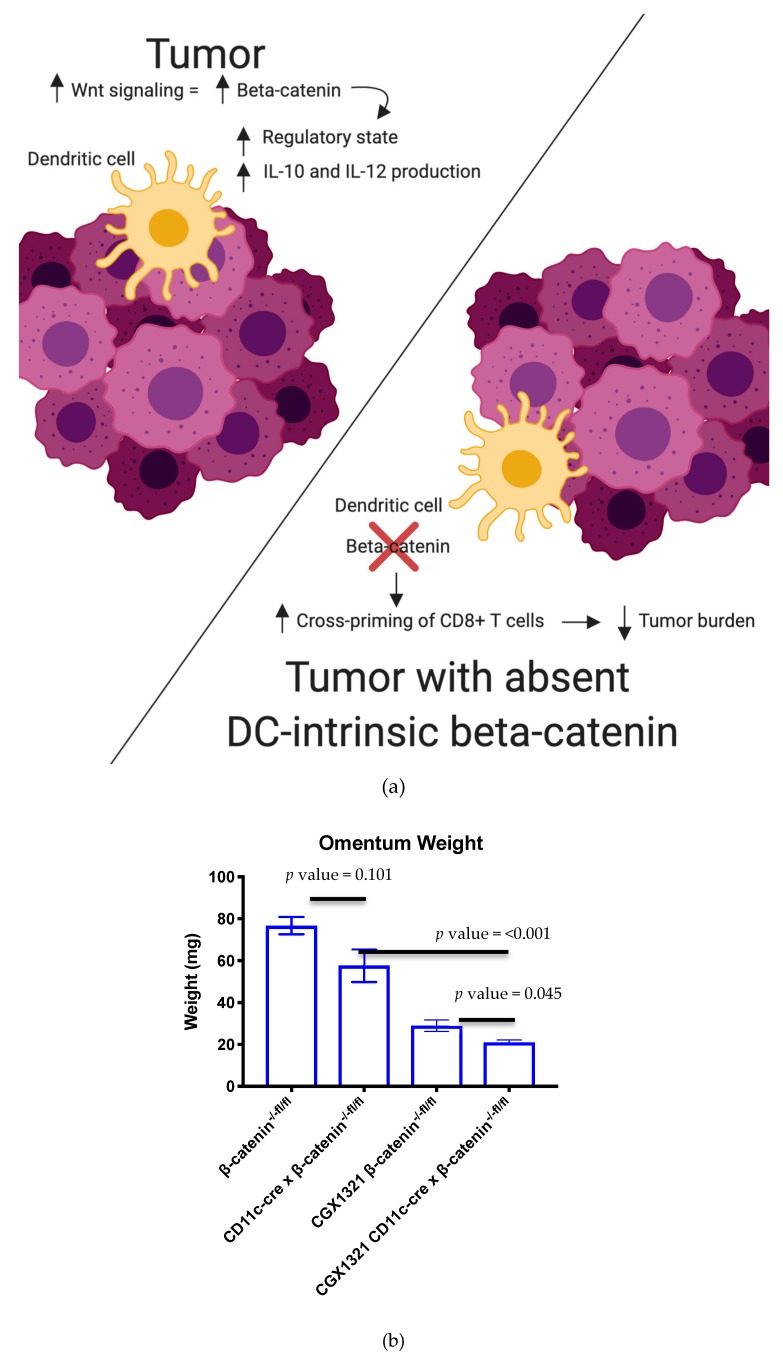
Inhibition of β–catenin signaling in dendritic cells decreases tumor burden. (**a**) Increased Wnt/β–catenin signaling is thought to increase IL-10 and IL-12 production and increase the regulatory state of DCs. The absence of intrinsic DC β–catenin may promote antigen priming of CD8^+^ T cells [36]. (**b**) Omentum weight decreased with loss of β–catenin in CD11c-cre × β-catenin^−/−fl/fl^ mice. Treatment with CGX1321 further increased this difference. (**c**) CD8^+^ T cells increased in the tumor microenvironment in CD11c-cre × β-catenin^−/−fl/fl^ mice with ID8 tumor challenge, a finding that was further exaggerated with CGX1321 treatment.

**Figure 7 cancers-12-00766-f007:**
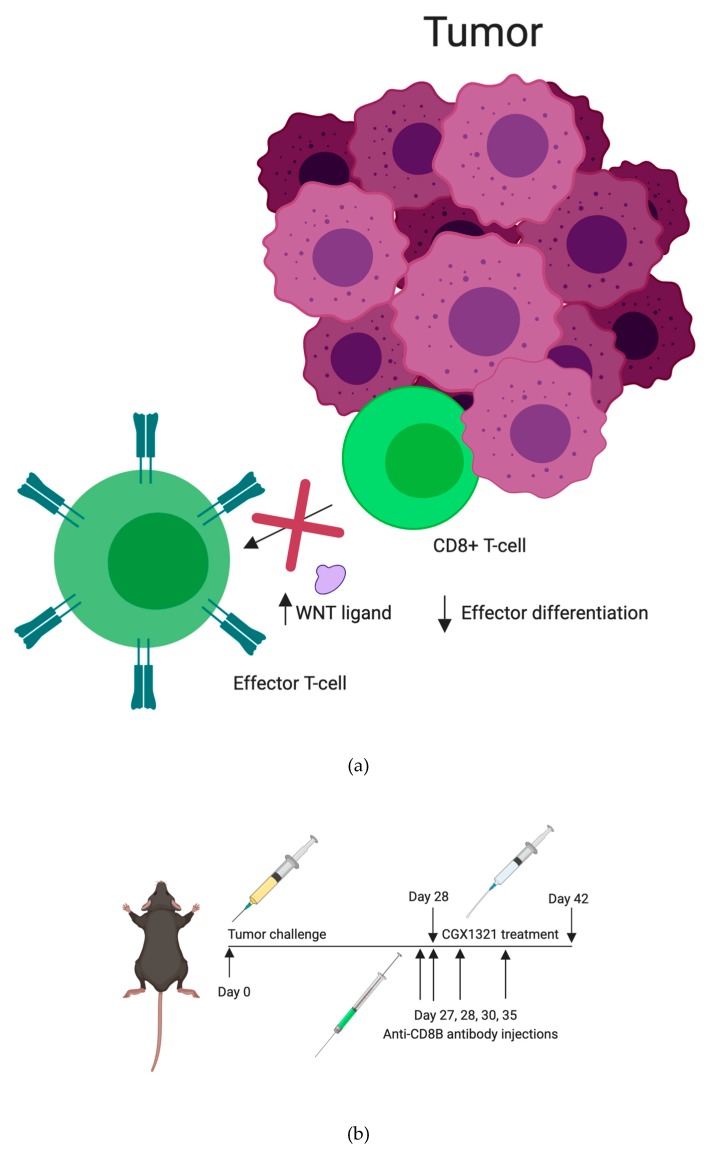
CD8^+^ T cells were required for Wnt inhibition to significantly inhibit tumor progression in ID8 in vivo model. (**a**) Graphical depiction of increased Wnt ligand leading to decreased CD8^+^ T cells effector differentiation in the TME. (**b**) Timeline of C57Bl/6 ID8 tumor challenge, with anti-CD8^+^β antibody injection, with or without CGX1321 treatment. (**c**) Tumor progression increased with CGX1321 treatment in the absence of CD8^+^ T cells in ID8 omentum.

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
