# Peer review of "Inhibiting WNT Ligand Production for Improved Immune Recognition in the Ovarian Tumor Microenvironment"

_cancers, 2020, doi:10.3390/cancers12030766_

Round 1

Reviewer 1 Report

The authors have responded well to the previous critique.

Author Response

We have corrected some grammatical errors. Thank you for your guidance. 

Reviewer 2 Report

Thanks for the responses to the previous questions.

Author Response

(The authors gave the same response as above.)

Reviewer 3 Report

I am mostly satisfied with the changes. However, I strongly encourage authors to create a table with RNA RIN numbers. specifically mentioning the samples with no RIN or low RIN numbers. 

Author Response

Thank you for this suggestion. We have added in the RIN values for samples as Figure S1c and addressed this more thoroughly in the RNA sequencing description.

This manuscript is a resubmission of an earlier submission. The following is a list of the peer review reports and author responses from that submission.

Round 1

Reviewer 1 Report

                Goldsberry et al. present data demonstrating that inhibition of Wnt signaling by the Porcupine (PORCN) inhibitor CGX1231, decreases growth of ovarian tumors in an ID8 mouse model.  They also demonstrate that decreasing WNT ligand production results in increases of T cell, macrophage and dendritic cell function. However, tumor growth was also reduced by the inhibitor in mice that lacked ß-catenin signaling in dendritic cells (DCs).  The authors hypothesize that decreasing Wnt ligand activity results in an activated T cell environment and ultimately improves survival in ovarian cancer. This is a clearly written paper that describes a complicated biological situation. The results may have implications for the design of new therapies in ovarian cancer.  However, revision in certain aspects of data presentation and analysis would strengthen the report.  Specific points include the following:

Abstract Please briefly describe ID8 tumors.

Introduction

Line 118 The authors state that the Trp53 gene is commonly mutated in human ovarian HGSC. Please provide the percentage of tumors that are p53 deficient here.

Results

Fig. 2A  A minority of the patients  in this study had a “cold” T cell signature.  Has this been noted previously?

Fig. 2B What was the pre- treatment status of the 4 samples that showed a decrease in Wnt activity? Were they “hot” or “cold”?

Fig. 3 Please provide a brief description of the ID8 p53 negative line.  In what other ways does it differ from the parental line?

Fig. 3 Please provide more details on the animal study.  For example, how many mice were used.  Was the study repeated?

Although CGX1321 increased survival in the ID8 parental line, it did not in the p53 negative line.  Since the majority of human HSC tumors are p53 negative, what implication might this have for the clinical utility of this inhibitor?

Line 180-182 The authors correctly noted that the ID8p53 negative tumors may have a decreased dependence on the Wnt signaling pathway. What other pathways might be important in this line?  Has this been previously described?

Fig. 5 The authors present very convincing flow cytometric data to illustrate changes in DCs, macrophages and T cells. Any histological data that might be available would be of interest.

Fig. 6B The authors find that knock-out of ß-catenin in DC cells results in decreased tumor weight using the ID8 model. The CGX inhibitor decreases tumor weight in control mice.  However, the CGX inhibitor also results in a decrease of tumor weight in the knock out animals.   This appears to be counterintuitive, but the authors suggest that this result is due to across priming of CTLs.  Do the authors have data on the percentages of CD8+T cells after treatment with the inhibitor?

Fig. 7 The authors need to show omentum weight from animals that only received the CD8 antibody.

Author Response

Review 1:

  Goldsberry et al. present data demonstrating that inhibition of Wnt signaling by the Porcupine (PORCN) inhibitor CGX1231, decreases growth of ovarian tumors in an ID8 mouse model.  They also demonstrate that decreasing WNT ligand production results in increases of T cell, macrophage and dendritic cell function. However, tumor growth was also reduced by the inhibitor in mice that lacked ß-catenin signaling in dendritic cells (DCs).  The authors hypothesize that decreasing Wnt ligand activity results in an activated T cell environment and ultimately improves survival in ovarian cancer. This is a clearly written paper that describes a complicated biological situation. The results may have implications for the design of new therapies in ovarian cancer.  However, revision in certain aspects of data presentation and analysis would strengthen the report.  Specific points include the following:

Abstract Please briefly describe ID8 tumors.

Thank you for this suggestion. A description has been added.

Introduction

Line 118 The authors state that the Trp53 gene is commonly mutated in human ovarian HGSC. Please provide the percentage of tumors that are p53 deficient here.

This percentage has been added and referenced.

Results

Fig. 2A  A minority of the patients  in this study had a “cold” T cell signature.  Has this been noted previously?

We defined cold, intermediate and hot as the bottom, middle and top tertile, so by definition 1/3 of the samples are cold.  We compared this with the Luke 2019 CCR paper that found that 34% of tumors across TCGA are cold and 34% were hot. In ovary specifically, they found about 25% were cold, which was consistent with our findings.

Fig. 2B What was the pre- treatment status of the 4 samples that showed a decrease in Wnt activity? Were they “hot” or “cold”?

For the 4 samples with the most decrease in Wnt signaling, 3 of the 4 are were initially “hot”, and 1 was “intermediate.” This has been clarified in the text.

Fig. 3 Please provide a brief description of the ID8 p53 negative line.  In what other ways does it differ from the parental line?

More details on the established differences between these two cells lines have been added under Results 2.3. Histologic comparisons have been added to Supplmental Material, Figure S2c.

Fig. 3 Please provide more details on the animal study.  For example, how many mice were used.  Was the study repeated?

We realized more details were needed in this section. Details on the experiments have been added to this section, such as number of mice and repeated experiments. Also, throughout the additional text we have tried to be more consistent about reporting these details.

Although CGX1321 increased survival in the ID8 parental line, it did not in the p53 negative line.  Since the majority of human HSC tumors are p53 negative, what implication might this have for the clinical utility of this inhibitor?

This is an excellent inquiry that is now further addressed in the Discussion section. From these data, thismutation may influence alterations of signaling pathways in this murine ovarian cancer model, but it is unclear how this is reflected in human HGSC.  Further investigation is warranted in additional cells lines, although this was beyond the scope of our current study.

Line 180-182 The authors correctly noted that the ID8p53 negative tumors may have a decreased dependence on the Wnt signaling pathway. What other pathways might be important in this line?  Has this been previously described?

The ID8p53-/-cell line is currently being studied. The literature has not shown other pathways that have been altered in this cell line yet, but we are eager to see future results.

Fig. 5 The authors present very convincing flow cytometric data to illustrate changes in DCs, macrophages and T cells. Any histological data that might be available would be of interest.

Histologic data was collected comparing untreated ID8 omentum tumors and IDp53-/-omentum tumors after 42 days of tumor challenge to illustrate the difference in aggressiveness between these two cell lines. These H&E slides have been added to the supplemental material, Figure S2c. However, histological data is not available for experiments after treatment. These data may be obtained with future experiments, but would not be available in the time frame outlined for this manuscript.

Fig. 6B The authors find that knock-out of ß-catenin in DC cells results in decreased tumor weight using the ID8 model. The CGX inhibitor decreases tumor weight in control mice.  However, the CGX inhibitor also results in a decrease of tumor weight in the knock out animals.   This appears to be counterintuitive, but the authors suggest that this result is due to across priming of CTLs.  Do the authors have data on the percentages of CD8+T cells after treatment with the inhibitor?

Thank you for bringing this key point to our attention. Flow cytometry was performed on ID8 omentum tumors in these mice after CGX1321 treatment as well. These data have now been added to Figure 6c. When cellular percentages were compared, there was a statistically significant increase in CD8+T cells with CGX1321 treatment, further supporting tumor recognition by these cells as a mechanism to decreased tumor burden. This has been more directly addressed in the text, Line 269, Line 381.

Fig. 7 The authors need to show omentum weight from animals that only received the CD8 antibody.

This data has been added to Figure 7 as Null Anti-CD8B Antibody and added to the supplemental figure, Figure S8.

Reviewer 2 Report

In this manuscript, the authors try to study the potential relationship of Wnt/b-catenin pathway and T cell signatures in determining ovarian cancer outcome. Their results showed the anti-tumor effect of a Wnt/b-catenin inhibitor CGX1321 in syngeneic ID8 ovarian cancer mouse model that associated with high Wnt/b-catenin expression. However, the potential relationship of Wnt/b-catenin and T cell signatures are not clear. Some of the data arrangement and explanation are quite confused.

Major points:

Figure 2: It is important to include the gene list or the calculation method of T cell signature in 2a. Please clarify how you group the cold, intermediate and hot tumors. What is the p-value in 1b? Figure 2: As the author declared that ‘decreased Wnt signaling could lead to increase TILs, which would improve treatment sensitivity and potentially overall survival (line 290-291)’, how about the survival curve of the patients in 2a according to the group method? How about the expression of wnt activity towards the NACT treatment response? Figure 4: Please clarify the purpose of this data, like may indicate the importance of b-catenin activation in determining the drug effect. Figure 5: Please highlight the phenotype of DC and macrophage. Normally an additional marker F4/80 will be used in determining macrophage. 5b, it was showed that the proportion of CD8+T cells in TME was decreased upon CGX1321 treatment (p=0.051). This result is controversial with the title and the main finding of the manuscript. TIL proportion is one of the most important standard for the hot tumor. Systematically drug treatment will affect the wnt/b-catenin pathway in most of the cells including tumor cell and immune cells. As Wnt/b-catenin is widely expressed by different cells, why the authors focused on DC intrinsic b-catenin pathway? Figure 6b: there was still anti-tumor effect of CGX1321 treatment in DC-b-catenin KO mouse, which indicate the potential role of wnt/b-catenin in other cells including tumor cells. CGX1321 treatment in ID8 cells with or without b-catenin will at lease clarify the role of tumor-intrinsic b-catenin together with DC-intrinsic ones. Figure 7c: CD8 T cell depletion did not influence on the anti-tumor effect of CGX1321, which is partially consistent with Figure 5b that the anti-tumor effect of CGX1321 may not be dependent on TIL, in which the functional mechanism need to be further clarified and not consistent with the title ‘ reverse T cell exclusion’.

Minor points:

Line 59, should be CD8+/Treg ratio instead of CD8/CD4 ratio Line 62, macrophage is one of the antigen-presenting cells. So the parallel expression of macrophage and APC is not accurate.

Author Response

Reviewer 2:

In this manuscript, the authors try to study the potential relationship of Wnt/b-catenin pathway and T cell signatures in determining ovarian cancer outcome. Their results showed the anti-tumor effect of a Wnt/b-catenin inhibitor CGX1321 in syngeneic ID8 ovarian cancer mouse model that associated with high Wnt/b-catenin expression. However, the potential relationship of Wnt/b-catenin and T cell signatures are not clear. Some of the data arrangement and explanation are quite confused.

Major points:

Figure 2: It is important to include the gene list or the calculation method of T cell signature in 2a. Please clarify how you group the cold, intermediate and hot tumors. What is the p-value in 1b?

A new table has been added to Supplemental Figures, Figure S1a, to include the genes we used to determine the T cell signature with the associated cell type.  We defined cold, intermediate and hot as the bottom, middle and top tertile, so by definition 1/3 of the samples are cold.  We compared this with the Luke 2019 CCR paper that found that 34% of tumors across TCGA are cold and 34% were hot. In ovary specifically, they found about 25% were cold, which was consistent with our findings. Previous Figure 2b has been replaced with what we hope is a more comprehensive graph.

Figure 2: As the author declared that ‘decreased Wnt signaling could lead to increase TILs, which would improve treatment sensitivity and potentially overall survival (line 290-291)’, how about the survival curve of the patients in 2a according to the group method? How about the expression of wnt activity towards the NACT treatment response?

Thank you for this strong recommendation. Kaplan-Meier curves have been added for PFS and OS in correlation to decreased Wnt activity. Furthermore, we have replaced the prior Figure 2B with a figure correlating PFS with fold change of Wnt pre vs post-NACT. We hope these changes clarify the improved patient outcome and benefit with decreased Wnt activity. Additionally, we illustrated the T cell signature pre vs post-NACT with the markers in Figure 2c.

Figure 4: Please clarify the purpose of this data, like may indicate the importance of b-catenin activation in determining the drug effect.

We have attempted to clarify the changes in Wnt gene signature and decreased β-catenin levels relating to the lack of response to CGX1321 in ID8p53-/-tumors.

Figure 5: Please highlight the phenotype of DC and macrophage. Normally an additional marker F4/80 will be used in determining macrophage. 5b, it wasshowed that the proportion of CD8+T cells in TME was decreased upon CGX1321 treatment (p=0.051). This result is controversial with the title and the main finding of the manuscript. TIL proportion is one of the most important standard for the hot tumor. Systematically drug treatment will affect the wnt/b-catenin pathway in most of the cells including tumor cell and immune cells. As Wnt/b-catenin is widely expressed by different cells, why the authors focused on DC intrinsic b-catenin pathway?

Thank you for this recommendation. The F4/80 marker was used as part of the staining in flow experiments for macrophage identification. This has now been further clarified in the materials section. Also, thank you for pointing out the discrepancy with CD8+T cells with CGX1321 treatment. We realize these results are confusing and have attempted to address it further throughout the text. Additionally, we have re-arranged the figure, adding the T-reg and CD8+T cell data into the Supplemental Materials section. We agree that this result was not consistent with the title. We have updated the title to be more reflective of the data, from “Inhibiting WNT ligand production to reverse T cell exclusion in the ovarian tumor microenvironment” to “Inhibiting WNT ligand production for improved recognition in the ovarian tumor microenvironment.” While the TME is complex, and there seems to be a combination of cells used in tumor recognition, we focused on DC specific β–catenin changes after our data suggested DCs were increased with tumor decrease with CGX1321 treatment, and DC function was increased on Nanostring data after treatment. Additionally, in other data, deletions of β-catenin in DCs were found to reduce Treg responses and delay tumor growth.  These indications have been further clarified in the text, Line 88.

Figure 6b: there was still anti-tumor effect of CGX1321 treatment in DC-b-catenin KO mouse, which indicate the potential role of wnt/b-catenin in other cells including tumor cells. CGX1321 treatment in ID8 cells with or without b-catenin will at lease clarify the role of tumor-intrinsic b-catenin together with DC-intrinsic ones.

Thank you for this important point. We have addressed this result more clearly in this section. Particularly, we have added the CD8+T cell flow cytometry results after CGX1321 treatment to Figure 6c. These results indicated CD8+T cells as an important factor in tumor recognition and decreased tumor burden. This is further discussed in the discussion section, Line 383.

Figure 7c: CD8 T cell depletion did not influence on the anti-tumor effect of CGX1321, which is partially consistent with Figure 5b that the anti-tumor effect of CGX1321 may not be dependent on TIL, in which the functional mechanism need to be further clarified and not consistent with the title ‘ reverse T cell exclusion’.

While the depletion of CD8+T cells was correlated with an increase in omentum tumor weight with CGX1321 treatment, this was not statistically significant. As you have pointed out, this result may indicate a partial reliance on CD8+T cells for tumor recognition, but these results indicate there is not a full reliance, consistent with the complexity of the TME. We agree that the functional mechanism needed to be further clarified. Thank you for pointing out the inconsistency with the title. We have updated the title to be more reflective of the data, from “Inhibiting WNT ligand production to reverse T cell exclusion in the ovarian tumor microenvironment” to “Inhibiting WNT ligand production for improved recognition in the ovarian tumor microenvironment.”

Minor points:

Line 59, should be CD8+/Treg ratio instead of CD8/CD4 ratio Line 62, macrophage is one of the antigen-presenting cells. So the parallel expression of macrophage and APC is not accurate.

Thank you for these points. These have been corrected.

Reviewer 3 Report

Goldsberry el al in a report titled “Inhibiting WNT ligand production to reverse T cell  exclusion in the ovarian tumor microenvironment" set out to test the possibility of `

Minor

Line 25-Expand DC, it comes here before it comes in line 29

Authors need to revisit text. Writing uniformity will increase the readability of the manuscript. The manuscript needs streamlining the information and authors should try to make it a cohesive document.

Major

RNA seq RIN numbers are between 2.8-8. Is this acceptable? The RIN number for RNA seq accepted value should be greater than 7.

Fig. 2b What is the mean survival time without NACT? How is this data normalized between pre and post NACT? Can authors show two matched sample types on the same graph for wnt activity and PFS?

Fig. 4 Need proper labeling. Western blot band size needs to be stated

Fig S3 need high-resolution fig. especially fig a

Fig 6b the texts are going on top of each other.

Overall, this is interesting research. However, my major concern is RNA quality and RIN number. Additionally, the writing is somewhat irregular and authors can improve uniformity.

Author Response

Reviewer 3:

Goldsberry el al in a report titled “Inhibiting WNT ligand production to reverse T cell  exclusion in the ovarian tumor microenvironment" set out to test the possibility of `

Minor

Line 25-Expand DC, it comes here before it comes in line 29

Thank you for noticing this. It has been corrected.

Authors need to revisit text. Writing uniformity will increase the readability of the manuscript. The manuscript needs streamlining the information and authors should try to make it a cohesive document.

Thank you for this observation and recommendation. We have made efforts to streamline the text for a more consistent flow.

Major

RNA seq RIN numbers are between 2.8-8. Is this acceptable? The RIN number for RNA seq accepted value should be greater than 7.

While the range is large, the median RIN is 7.85, with half of the values being above this, and only 4 of the samples are below 5. RIN scores are not available for all tissues.  This median RIN value has been added to the text for clarification. This is now detailed in the Methods section.

Fig. 2b What is the mean survival time without NACT? How is this data normalized between pre and post NACT? Can authors show two matched sample types on the same graph for wnt activity and PFS?

Thank you for this excellent suggestion. We have now re-done this figure, as Figure 2d. We have added the T cell signatures pre-NACT and post-NACT as Figure S1d. From this, we used markers to clarify T cell signature pre-NACT vs post-NACT. Kaplan-Meier curves have also been added as Figure 2b and Figure 2c for PFS and OS. To calculate Wnt activity, we used the 37-gene signature from Luke et al. This is a median gene expression level for 37 wnt-relevant genes in ovarian cancer. Gene expression in general is normalized using the variant stabilization transformation in DESeq2. Then to look at the change we looked at the fold-change between post/pre by just dividing post by pre median Wnt activity score. These gene signatures have been added as Figure S1b.

Fig. 4 Need proper labeling. Western blot band size needs to be stated

Thank you for point this out. The comprehensive Western blot has been added as Supplemental Figure S3a, with the molecular weight ladder. Additionally, the band size has been added to the figure. 

Fig S3 need high-resolution fig. especially fig a

Thank you. New images have been uploaded to replace the prior figures.

Fig 6b the texts are going on top of each other.

This may have been due to a formatting error. Thank you for noticing this. It should be corrected.

Overall, this is interesting research. However, my major concern is RNA quality and RIN number. Additionally, the writing is somewhat irregular and authors can improve uniformity.

Thank you for your comments. We have made efforts to improve the quality of the uniformity of the writing content. We have addressed the RIN and RNA quality in the methods section, in hope of clarification.